# Polysaccharides from the Sargassum and Brown Algae Genus: Extraction, Purification, and Their Potential Therapeutic Applications

**DOI:** 10.3390/plants12132445

**Published:** 2023-06-25

**Authors:** Elda A. Flores-Contreras, Rafael G. Araújo, Arath A. Rodríguez-Aguayo, Muriel Guzmán-Román, Jesús Carlos García-Venegas, Erik Francisco Nájera-Martínez, Juan Eduardo Sosa-Hernández, Hafiz M. N. Iqbal, Elda M. Melchor-Martínez, Roberto Parra-Saldivar

**Affiliations:** 1Tecnologico de Monterrey, School of Engineering and Sciences, Monterrey 64849, Mexico; eldafc@tec.mx (E.A.F.-C.); rafael.araujo@tec.mx (R.G.A.); a01209135@tec.mx (A.A.R.-A.); a01658784@tec.mx (M.G.-R.); a01383937@tec.mx (J.C.G.-V.); a00832573@tec.mx (E.F.N.-M.); eduardo.sosa@tec.mx (J.E.S.-H.); hafiz.iqbal@tec.mx (H.M.N.I.); 2Tecnologico de Monterrey, Institute of Advanced Materials for Sustainable Manufacturing, Monterrey 64849, Mexico

**Keywords:** brown macroalgae, bioactivity, biopolymer, extraction methods, fucoidan, laminarin, alginates

## Abstract

Brown macroalgae represent one of the most proliferative groups of living organisms in aquatic environments. Due to their abundance, they often cause problems in aquatic and terrestrial ecosystems, resulting in health problems in humans and the death of various aquatic species. To resolve this, the application of *Sargassum* has been sought in different research areas, such as food, pharmaceuticals, and cosmetics, since *Sargassum* is an easy target for study and simple to obtain. In addition, its high content of biocompounds, such as polysaccharides, phenols, and amino acids, among others, has attracted attention. One of the valuable components of brown macroalgae is their polysaccharides, which present interesting bioactivities, such as antiviral, antimicrobial, and antitumoral, among others. There is a wide variety of methods of extraction currently used to obtain these polysaccharides, such as supercritical fluid extraction (SFE), pressurized liquid extraction (PLE), subcritical water extraction (SCWE), ultrasound-assisted extraction (UAE), enzyme-assisted extraction (EAE), and microwave-assisted extraction (MAE). Therefore, this work covers the most current information on the methods of extraction, as well as the purification used to obtain a polysaccharide from *Sargassum* that is able to be utilized as alginates, fucoidans, and laminarins. In addition, a compilation of bioactivities involving brown algae polysaccharides in in vivo and in vitro studies is also presented, along with challenges in the research and marketing of *Sargassum*-based products that are commercially available.

## 1. Introduction

The seaweed problem is a global environmental issue that has recently been receiving attention due to the inundation of beaches by seaweed. This problem is caused by Ulva, a green seaweed, and a golden/brown floating or pelagic seaweed, Sargassum, causing a change in the color of the tides, observed as gold, mainly in West Africa and the Caribbean [1]. The currents that bring these seaweeds to the coastlines begin in the Sargasso Sea, a region with a strong presence of these algae; the Sargasso Sea is considered the main source of pelagic Sargassum. In addition, it has been considered that this excessive increase in Sargassum may be due to the large amount of nutrients in the ocean that come from the sand of the Sahara Desert and from the deforestation of the Amazon in the Brazilian basins. There is also the hypothesis that the lack of cyclones in the last few decades has prevented the dispersion of Sargasso, allowing it to concentrate in certain regions [2]. Sargassum clogs up beaches, which harms marine life by blocking sunlight from reaching the algae and seagrass below it. It includes economic problems, such as reduction of tourism and alteration of local fishing activities [2].

To solve the problem of Sargassum, different applications have been sought, such as using it as a source of food, extracting its health-related biomolecules, finding agricultural uses for it, utilizing it in the bioremediation of effluents, taking advantage of it as a clean energy source, or extracting molecules from it, polysaccharides being the most important [3].

The large extensions of the macroalgae can capture large amounts of CO_2_, most of which are converted to polysaccharides, which play a key role in carbon cycling [4]. The most important polysaccharides are mainly laminarins, alginates, carrageenan, and fucoidans [5,6]. These have high potential for biological applications in pharmaceutical products, cosmeceuticals, and functional foods; additionally, its structure and composition are determined by the algae species, but these can be influenced by other factors causing variation in the formation of these polysaccharides.

Macroalgae have high fiber content attributed to the non-digestible polysaccharides in the cell wall. To obtain such biocompounds, it is essential to find the most efficient extraction method and optimize the key parameters for a better yield and extraction composition [7]. Due to the disadvantages brought by the traditional methods that caused organic and environmental pollution, new techniques were developed with a better outcome overall. These new methods are faster, more efficient, and sustainably extract biocompounds from natural resources. Modern methods, such as supercritical fluid extraction (SFE), pressurized liquid extraction (PLE), subcritical water extraction (SCWE), ultrasound-assisted extraction (UAE), enzyme-assisted extraction (EAE), and microwave-assisted extraction (MAE) are now used for the extraction of specific compounds [8].

As mentioned before, marine algae are rich in nutrients and biocompounds [9]. Phycocolloids are polysaccharides derived from seaweeds and have very diverse physicochemical characteristics [10]. The phycocolloids found in *Sargassum* sp. are fucoidan, alginate, and laminarin [11]. They are believed to attribute different biological activities, such as neuroprotective effects [12], antioxidant activities [13], antitumor potentials, anti-collagenase activity, antimicrobial effects, and more [14].

Fucoidans are claimed to be the major bioactive compound found in *Sargassum* sp.; its main monomer is the fucose. The chemical structure of fucoidans depends on several factors, such as the species, geographical location of recovery, climatic conditions, etc. Extracting the fucoidan impacts its biological activities, but these activities also depend on their degree of sulphation, structural formations, and weight. Fucoidans can have antibacterial properties, are antiviral, contain antioxidants, and have anti-cancer and antitumor properties [11]. Some marine sources of fucoidan from brown algae are mozuku, kombu, limu moui, bladderwrack, and wakame [12].

Alginates are alginic acid salts and their derivatives. Their predominate function is to provide structure as a cell wall component due to their physicochemical properties, such as gel formation and viscosity [12]. Studies have demonstrated that alginic acid prevents the absorption of heavy metals in the body. It has also been proven that some derivates from alginate act as potential preventive neurodegeneration biocompounds. In addition, alginic acid reduces cholesterol and plays an important role as a dietary fiber, which makes it beneficial for health [15]. Some of the sources of alginate are *Laminaria hyperborea*, *L. digitata*, *Macrocystis pyrifera*, *Ascophyllum nodosum*, and *L. japonica* [16].

The phycocolloid laminarin is a biodegradable and non-toxic linear polysaccharide and the storage carbohydrate in brown algae it is extracted from the cell wall of different species, such as *Laminariaceae*, *Laminaria*, *Saccharina*, *Eisenia*, or *Fucus*. This biocompound has the capacity to act as an antitumor, antioxidant, and anti-inflammatory agent, and also has prebiotic properties [17]. Researchers have shown that some chemical modifications and processing alterations can enhance laminarin’s bioactivity and therapeutic properties, such as anti-inflammatory, anti-apoptotic, antitumor, antioxidant, and anticoagulant activities. Even though some strategies have resulted successfully, difficulties have been encountered during various modification attempts [18].

Due to the physicochemical properties that are present within the biocompounds extracted from different sources of brown seaweeds, there exists a wide range of industries that are more and more interested in the application of them in various products, such as the food industry, pharmaceutical, cosmetics, and paint, each one being unique and versatile [9].

In recent years, biocompounds of marine sources have caught the most attention from the pharmacological industry, which has discovered an anticancer purpose for prostate cancer and compounds that are helpful in preventing osteoporosis; an antidiabetic treatment is also being investigated [19]. The food industry is one of the largest, and edible seaweeds have offered the capacity to develop functional foods for many years. Researchers have made a great effort in the past 15 years to discover new ways to include bioactive compounds in different meat products with the purpose of improving their nutritional value [20]. There are other types of industries, such as aquaculture, in which it is possible to use the biocompounds effectively to prevent diseases. The cosmetic industry has also used marine biocompounds as viscosifiers, stabilizers, and gelling agents [21].

Taking these numerous benefits into account, the purpose of this review is to collect recent literature about the methods of extraction and purification of polysaccharides from brown algae by means of the most innovative technologies, as well as to determine their different functionalities and applications as natural therapeutic agents that are currently available on the market.

## 2. Advanced Methods for Extraction and Purification of Brown Seaweed Polysaccharides

In recent years, the extraction process of polymers has transitioned from conventional methods to new techniques, due to the time- and energy-consuming practices of older methodologies. These emerging extraction methods have an impact on the biocompounds’ properties and biological activities, effectively improving the extraction process overall, Figure 1a. This is achieved by optimizing the extraction parameters, such as time, power, temperature, solvent, etc., and developing pre-treatments depending on the matrix of use and the conditions that are presented [22]. The aim of using these recent methodologies is to obtain a higher yield and better quality of the bioactive compounds of interest [23]. For potential industrial applications, it is also important to maintain the structural integrity and beneficial properties of the polymers [24,25].

Technologies such as microwave-assisted extraction (MAE) work via a dipole rotation of a polar solvent due to the radiation conducted by the dissolved ions. Ultrasound-assisted extraction (UAE) is based on cavitation, sound waves, and heat and enzyme-assisted extraction (EAE) utilizes enzyme hydrolysis, which breaks the bonds and liberates the biocompounds; both of these methods are considered non-traditional techniques [26]. These three are the most reported techniques, although there are plenty of others, such as negative pressure cavitation (NPC); hydro-diffusion extraction (HDE); supercritical fluid extraction (SFE); subcritical water extraction (SCWE) [27]; or combined methods, such as enzyme-ultrasonic-assisted extraction (EUAE) and microwave-assisted aqueous two-phase extraction (MAATPE). Several treatments have been tested, but unfortunately, a universal extraction method has not yet been selected [28].

As mentioned before, biocompound extraction involves several steps (sample preparation, pre-treatment, extraction, polymer recovery, and purification) and optimization of the parameters [23]. The most recent parameters of polysaccharides extraction from brown seaweed are shown in Table 1.

Fucoidan MAE extraction with *Sargassum siliquosum* provided a yield of 6.94% under the conditions of 750 W, 10 min, and 15 mL/g with EtOH as a solvent [36]. While using *Fucus* sp. as a matrix, the optimal conditions were a 30 min treatment at 120 °C with sulfuric acid as a solvent, resulting in a 5.56% yield [30]. It should be noted that investigations about alginate UAE extraction have reported a 27% recovery with *Sargassum binderi* by adjusting the parameters to 150 W for 30 min at 90 °C, 25 kHz in a water solvent, and controlled pH [35]. Ultrasound extraction has also proven successful while applying 1.5 A, 50 W, and 40 Hz for 30 min at room temperature with water as a solvent; this provides a 15% yield [34].

The purification of polysaccharides is important to understand the sulfated variations of chemical structures of the different species from which these biocompounds are obtained; this also depends on the extraction methods, harvesting time, and the location of the algae, Figure 1b [37].

The most researched methods for purifying polysaccharides are anion-exchange chromatography [38], DEAE cellulose (diethylaminoethyl-cellulose) ion-exchange column chromatography [39], lyophilization, and ultrafiltration combined with IEX (ion-exchange chromatography) [40]. Anion-exchange chromatography can separate a large range of molecules by segregating them based on their net surface charge. Chromatography and its variations are the most common methods to purify biocompounds. Lyophilization or freeze-drying consists of eliminating the water from the sample of interest by freezing the water; afterwards, the frozen water is removed by a vacuum, bypassing the matter from the solid state to vapor (via sublimation), and the unfrozen water is eliminated by desorption [41]. Ultrafiltration is the separation of materials, such as fats, particles, and proteins, among others, of the sample of interest through the use of membranes that can have a pore size of 0.01 or 0.1 µm, as well as pressure ranging from 3.4 to 8.3 bar [42]. On the other hand, in the dialysis purification technique, the solution to be dialyzed is immersed in a specific buffer and placed in a sealed semi-permeable membrane, which allows the passage of small molecules through diffusion, balancing the compounds of interest and the dialysate. This is accomplished through the selective permeability of the membrane, which allows small molecules and buffer salts to pass through, resulting in an alteration of the concentration of salt and the molecule of interest in the dialyzed solution [43].

These methods are commonly used to purify polysaccharides, fucoidans, proteases, and alginates. These products are obtained by different extraction methods, and they are derived mostly from brown algae and the different species of *Sargassum*. Polysaccharides of great interest because of their biological properties, since they act as anti-lipogenic, antioxidant, anticoagulant, anti-inflammatory, antiviral, and antitumor. The bioactivities of fucoidan are influenced due to its structural features, such as its chemical composition, molecular weight, and degree of sulfation [38].

The purification methods and conditions of different species of *Sargassum*, as well as the yield obtained are summarized in Table 2. Some methods seem to have obtained high yields of purification. A study based on the purification of *Sargassum ilicifolium* [44] obtained a maximum yield of 8 ± 0.9%. This study used different extraction methods, showing better results with the sonication–microwave method than the hot water method.

The extraction method makes a difference in the yield of the purified products obtained from the *Sargassum*; one example is the aforementioned study of *Sargassum ilicifolium*. However, there are more conditions that may change the purification yield, such as the extraction times and temperatures, which can be observed in the study of *Sargassum polycystum*: this study was based on the extraction of different polysaccharides, such as fucoidans, alginates, and laminarin [45]. The final application of the bioactive compounds extracted from macroalgae will define the extraction and purification strategies, since the characteristics, such as the degree of purity and properties or bioactivities of the compounds of interest, are defined in each treatment, which can decrease or increase the specific bioactive potential of the compounds.

**Table 2 plants-12-02445-t002:** Polysaccharide purification methods from brown macroalgae.

Algae	Purification Method	Purification Conditions	Yield	Reference
*Sargassum ilicifolium*	Chromatography	Dissolved in 5 mL distilled water. Loaded to a pre-equilibrated DEAE cellulose 52 columns. The stepwise gradient elution with 0.05 M Tris–HCl buffer (pH 7.0) containing 0.5, 2.0, 3.5, and 5.0 M NaCl. Fractions of 4 mL per tube were collected and monitored at 490 nm by the phenol–sulfuric acid (H_2_SO_4_) method.	Fucoidan yield from sonication–microwave extraction: 8 ± 0.9%. Hot water extraction: 6 ± 0.5%.	[44]
*Sargassum autumnale*	Chromatography	Twice the volume of 99.5% ethanol solution was added to the enzyme-assisted hydrolysates precipitate and collected by centrifugation.	Different extraction methods: Distilled water extract: 17.23 ± 0.28. Ultraflo extract: 23.50 ± 0.56. Protamex extract: 23.89 ± 0.42.	[46]
*Sargassum aquifolium*	Isopropyl alcohol purification process	Not reported	21.74 ± 2135	[47]
*Sargassum patens*	Dialysis	Pr: (NH_4_)_2_SO_4_ 85% Pu: DI (kDa n.s.)	8.2%	[38]
*Sargassum polycystum*	Ultrafiltration	Not reported	7.27%	[48]
*Sargassum siliquosum*	Anion-exchange chromatography	Protein and uronic acid were removed.	5.08 ± 1.17%	[49]
*Sargassum natans*	Gel permeation chromatography	Not reported	Maximum of 21.21%	[13]
*Sargassum fusiforme*	Lyophilization	Not reported	On the sample ASFF: 11.24 ± 0.94a	[50]
*Sargassum swartzii*	High-performance liquid chromatography	A total of 1 mg of lyophilized SP was dissolved in 1 mL distilled water. Filtered through a 0.22 μm syringe tip filter and subjected to HPLC analysis.	Hot water extraction: 3.6% HCl method: 1.2%	[45]

## 3. Structural Description of the Main Polysaccharides Found in Brown Macroalgae

Brown algae are a source of protein, fatty acids, carbohydrates, minerals, and vitamins (Figure 2). Polysaccharides are the main component, with a range of 50–60% dry water (DW), which functions as storage energy [22,51,52]. Polysaccharides are mainly made up of mannose, glucose, galactose, and fucose, along with small proportions of arabinose and rhamnose. Brown algae are mainly composed of three polysaccharides, which are fucoidans, alginate, and laminarin, but their conformation varies depending on the species of brown algae. These variations in the content of polysaccharides are due to conditions such as population age, temperature, and geographic location, and this is reflected in their biological activities that allow them to act as anti-inflammatory, anticoagulant, antiviral, or antioxidant agents, among others [22,52].

### 3.1. Fucoidans

Fucoidans are anionic sulfated polysaccharides located in the intracellular space and cell wall. Fucoidans are mainly made up of repeating units of L-fucose (α (1→3)-L-fucopyranose residues, alternating α (1→3) and α (1→4)-linked L-fucopyranosyls, or both forms, and the sulfate groups that correspond to 5% to 38% depend on the species of brown algae. Fucoidans are also composed of uronic acid, acetyl groups, proteins, and other monosaccharides, such as galactose, mannose, glucose, rhamnose, and xylose. These polysaccharides span molecular weights from 7 to 2300 kDa, corresponding to 4 to 10% of the DW of brown algae. Being a very heterogeneous polysaccharide, it presents a diversity of structures. It has been observed that its biological activity is correlated with the molecular weight, the monosaccharides that make it up, and its chemical composition, sulfation degree, position of the sulfate groups, and glycosidic bonds [22,52,53,54].

### 3.2. Alginate

Another polysaccharide of great interest is alginate, which is located in the cell wall and in the intracellular matrix of the different species of brown algae and represents 40% of the DW. Alginate is a linear polysaccharide mostly made up of α-L-guluronic acid (G) and β-D-mannuronic acid (M) isomeric residues, which are linked by the 1,4-glycosidic configuration, giving rise to three types of arrangements, namely, GG, MM, and GM, depending on the proportion. This configuration allows the alginate to increase its viscosity and gelling properties; for example, the viscosity is assigned by the M blocks, and the gelling properties are provided by the G blocks. This type of block presents greater solubility in water than the M blocks. These properties can also be increased with Ca^2+^ ions [22,52,53,54].

### 3.3. Laminarin

Laminarin is a polysaccharide found in cell vacuoles. It represents 22 to 49% of DW and serves as an energy reserve in brown algae. In addition, it has a high concentration of neutral sugars, and a low amount of uronic acid. This polysaccharide of low molecular weight (approx. 5 kDa) is soluble in water and is made up of β-glucan consisting of β-1,3-D-glucopyranose residues, linked by β-1,6-intrachain unions and at their ends. Reduced laminarin may have residues of glucose or mannitol; the degree of polymerization is between 20–25 moieties of glucose. This polysaccharide has various biological properties, such as the ability to stimulate an antitumor response, promoting wound repair, and enhancing the activity of the immune system [22,52,55].

## 4. Exploring the Potential Relationship between Polysaccharides Structures and Their Bioactivities

Brown macroalgae polysaccharides are known for their several potential therapeutic properties; in fact, they are used as an ingredient or component in a wide range of industries, including pharmaceutical, medical, food, and cosmetics [54]. The most promising activities are in the field of medicine due to their antiviral, anti-inflammatory, antioxidant, and anticarcinogenic actions, Figure 3 [52].

### 4.1. Anticancer Activity

Cancer is considered the main cause of death worldwide; in 2021, more than 10 million people died from this disease. Cancer is defined as a malignant tumor or neoplasm of abnormal tissue mass, which has the potential to metastasize and attack any part of the body with a high risk of death [56]. Therefore, research and science have been focused on developing precise and effective alternative techniques to reduce cancer’s impact on health and improve conventional treatments. New technologies using nanomedicine and biomaterials are under evaluation in clinical trials and others are already in clinical practice [57]. Biocompatible materials have been promising elements because they can be bioengineered in different forms as nanoparticles with important advantages, such as selectivity and efficacy in the attack on tumor cells [58].

Several researchers around the world have been promoting and reporting the anticancer activity of brown algae and the behavior of cancer cell lines in in vitro assays. One study reported the anticancer activity of laminarin by using *L. japonica* at a concentration of 35 mg/mL to significantly decrease the Bel-7404 (human hepatoma cell line) viability; for 48 h, the viability was only 46.20%, and for HepG2, it was only 42.85%. Regarding the apoptosis rate for the Bel-7404 cell line, it was 2.72 higher with laminarin, and for HepG2, it was 8.18 times higher than without treatment (Table 3) [59].

The bioactivity of fucoidan has been widely used due to its inherent anticancer properties, favorable drug delivery behavior, and promising targeting ability, resulting in the induction of cell apoptosis and inhibition of angiogenesis [60]. For example, in some species of brown seaweed, such as *Turbinaria conoides*, the anticancer effect on the hepatoblastoma-derived (HepG2) cell line has been studied through a cell viability assay, using 3-(4,5-dimethylthiazol-2-yl)-2,5-diphenyltetrazolium bromide MTT and different concentrations (0–200 μg/mL) of fucoidan/quercetin treatments for 48 h. The results show that the fucoidan/quercetin treatment reduced cell viability to less than 50% in a concentration-dependent manner. A concentration of 200 μg/mL demonstrated a better performance. The results conclude that fucoidan had more significant anticancer activity compared to quercetin [61].

On the other hand, alginate has not been as widely used in anticancer activity as other brown algae polysaccharides, even though some reports of the cytotoxic effects alone or in combination with different compounds have demonstrated the potential that alginate could contribute to cancer treatments. A study using polysaccharides from brown algae *C. Sinuosa* showed the bioactivity of alginate by promoting a significant decrease in the cell viability of HCT-116 cells, with an IC50 of 690 μg/mL−1 and a 37.1% inhibition rate at 750 μg/mL^−1^. Nevertheless, fucoidan at similar high concentrations resulted in a better inhibition rate of 45% [62]. Alginate can play a major role in encapsulation as alginate-based hydrogels for cancer therapy, which can control and target drug administration, improving the stability and minimizing unwanted effects, time, and effort [63,64]. Recent research has reported that low-weight alginate oligosaccharides have better anticancer activity than complete polysaccharide because they prevent cancer cell proliferation and reduce tumor metastasis, in addition to providing antioxidant and anti-inflammation properties [65].

**Table 3 plants-12-02445-t003:** Biomedicine and therapeutic applications of polysaccharides from brown algae species.

Brown Algae	Polysaccharide	Bioactivity	Bioassay	Bioactivity Results	Reference
*L. japonica*	Laminarin	Anticancer	Cell viability detected by WST-8 cell proliferation assay, flow cytometry in 96-well plate in human HCC cell lines, including Bel-7404 and HepG2, when incubated with different concentrations of laminarin. Hepa 1–6 tumor-bearing mice were injected with different concentrations and tumors were measured.	Cell viability; ;aminarin concentrations of 35 mg/mL significantly decreased the Bel-7404 viability, with only 46.20% at 48 h. HepG2 was only 42.85% of that of cells without treatment. Apoptosis rate of Bel-7404 was 2.72 higher with laminarine and 8.18 times higher for HepG2 than without treatment. Tumor growth inhibition was higher at 1200 mg/k.d of laminarin with 67.92%.	[59]
*Padina pavonioca*	Sulfated polysaccharides	Anticancer Antioxidant	DPPH Cell viability: MTT assay cytotoxic activity in HeLa cancer cell lines cultured in DMEM supplemented.	A total of 1 mg/mL increases scavenging activity up to 63%. Low doses (0.05–0.1 mg/mL) exhibit cytotoxic activity in HeLa cancer cell line.	[66]
*Sargassum ilicifolium*	Fucoidan	Antioxidant Osteogenic ability (bone regeneration)	DPPH expression of osteoblast differentiation media in DMEM supplemented with murine mesenchymal stem cells (C3H10T1/2).	IC_50_ 0.96 mg/mL (crude) and 2.51 mg/mL (purified). A total of 1 μg/mL of purified fucoidan provides cell proliferation (130%) on C3H10T1/2 at 48 h.	[38]
*Silvetia Compressa* *Ecklonia arborea*	Sulfated polysaccharides (fucoidan, laminarin, alginate) and phlorotannins	Antioxidant	DPPH & ORAC	DPPH IC_50_ (mg/mL) *S.Compressa* 1.7 *E. arborea* 3.7 ORAC (mmol Trolox equivalent/g) *S.Compressa* 0.817 *E. arborea* 0.801.	[67]
*Sargassum siliquosum*	Fucoidan	Antioxidant Anti-inflammatory	DPPH (absorbance of 50% methanol solution mixed with the sample solution was the blank). Cell viability: RAW264.7 cell line in 24-well plate. TNF-α content level after lipopolysaccharide LPS exposure reflected the anti-inflammatory activity. Quercetin was used as a positive control.	Antioxidant results: EC_50_ of purified fucoidan 2.58 mg/mL, higher antioxidant ability showed crude extract with an EC_50_ of 0.34 mg/mL. Cell Viability: Inhibition of TNF-α reached 14.8% with 0.25 µg/mL of fucoidan-treated compared to LPS control.	[36]
*Sargassum horneri*	Alginic acid	Anti-Inflammatory	Cell culture RAW 264.7 mouse macrophages and HaCaT (human keratinocytes) cultured in DMEM, 10% FBS, and 1% antibiotics. In 24-well plates, HaCaT were seeded with SHA and Chinnese fine dust CFD at 125 µg/mL under optimized conditions. MTT assay was used for cell viability.	Cell viability: A concentration of SHA 25–100 µg/mL decreased viability of HaCaT in a range of 20%.	[37]
*Sargassum horneri*	Fucoidan	Anti-inflammatory	Cell culture and viability assay RAW 264.7 macrophages were cultured in DMEM, 10% FBS, and 1% antibiotics. MTT test in 24-well plate. Negative control had untreated macrophages, positive control was only treated with PBS.	Fucoidan in concentrations of 12.5–50 μg/mL inhibited the production in LPS-activated RAW 264.7 macrophages with IC_50_ = 40 µg/mL.	[68]
*Sargassum fulvellum*	Sulfated polysaccharides	Anti-inflammatory	Anti-inflammatory activity in RAW 264.7 macrophages cultured in DMEM medium. Stress induced by *E. coli* lipopolysaccharides. In vivo tests applied in zebrafish embryos to measure their survival rate after 3 days. Levels of ROS, heartbeat, and cell death after stress induced by *E. coli* lipopolysaccharides.	Viability of RAW 264.7 cells increased by 94.6%, while the production of nitric oxide (NO) in RAW 264.7 decreased by 40.7%. Zebrafish survival, cell death, and ROS/NO production decreased in a dose-dependent manner.	[29]
*Turbinaria decurrens*	Fucoidan	Anti-inflammatory	Swiss albino mice were subjected to formalin-induced paw edema. The mice were treated with the extracted fucoidan, which was administered orally, to evaluate the anti-inflammatory effect.	The mice treated with fucoidan showed reduction in the perception of the wound by the mice. The licking time was reduced by more than 50%. In addition, the anti-inflammatory effect of the fucoidan in paw edema showed a reduction of 52%.	[69]
*Sargassum fusiforme*	Sulfated polysaccharides	Anti-inflammatory	Sulfated polysaccharides extracted by an enzymatic method were tested in RAW 264.7 cells stressed with lipopolysaccharide (LPS). To study the anti-inflammatory effects of the polysaccharides, the level of expression of NO and inflammatory cytokines, such as TNF-α, IL-1β, IL-6, and PGE2, were determined.	The sulfated polysaccharides showed an anti-inflammatory activity with a dose-dependent behavior. The addition of the polysaccharides successfully reduced the expression of TNF-α, IL-1β, IL-6, and PGE2. Additionally, the production of NO was reduced, while the cellular viability increased.	[70]
*L. japonica*	Polysaccharides	Antiviral	The polysaccharides isolated by ethanol precipitation were tested in HEK293 cells infected with the respiratory syncytial virus (RSV) to study the antiviral activity of the polysaccharides. The expression of IRF3 and IFN-α were analyzed.	The polysaccharide extracts demonstrated significant antiviral activity against RSV by increasing IFN-α expression via regulation of the IRF3 signaling pathway in HEK293 cells.	[71]
*Sargassum polycystum*	Fucoidan fraction-2 (Fu-F2)	Antibacterial	The MIC and MBC were determined for bacterial strains *Streptococcus mutans*, *Staphylococcus aureus*, *Pseudomonas aeruginosa*, and *E. coli*. A 48-well microtiter plate was used. The antibacterial activity was determined by disk diffusion assay. Test bacteria were grown on Luria-Bertani agar medium and a crude fucoidan loaded disk was placed with the standard antibiotic disk (tetracycline).	The highest antibacterial activity (21 ± 1.0 mm) was obtained at 50 µg/mL against *Pseudomonas aeruginosa*, and the lowest activity (16 ± 0.53 mm) was against *Staphylococcus aureus*.	[72]
*Fucus vesiculosus*	Fucoidan	Anti-angiogenesis	Use of crude extracted fucoidan over endothelial cells and chicken embryos.	A concentration of fucoidan at 0.5 mg/mL was able to prevent the formation of tubular structures in epithelial cells. Chicken embryos presented a reduction in blood vessel formation, as well as in the tumoral mass.	[73]
*Fucus distichus subsp. evanescens*	Fucoidan	Anti-angiogenesis	Measurement of the gene expression of the angiopoietins 1 and 2, vascular endothelial growth factors, and stromal-derived factors in mono- and co-cultured systems of human outgrowth endothelial and human mesenchymal stem cells. Cells were treated for seven days with extracts obtained enzymatically from *Fucus distichus* subsp. evanescence. The anti-angiogenic activity of co-cultured cells was analyzed by measuring the length and area of the tube-like structures created by the endothelial and mesenchymal cells.	In monoculture: The fucoidan extract downregulates the expression of vascular endothelial growth factor and stromal-derived factor-1 in mesenchymal stem cells; however, the angiopoietins-1 and angiopoietins-2 in the outgrowth endothelial cells’ levels were not affected by the fucoidan extract. The fucoidan extract with the higher sulfate content was able to disturb the formation of the tube-like structures; length and area were both reduced.	[74]
*Ascophyllum nodosum*	Sodium alginate	Prebiotic	Alg-MAE (microwave-assisted extraction) *L. delbruecki* ssp. *bulgaricus* and *L. Casei* growth media. Inulin (positive prebiotic control), glucose (the negative control in a 96-well-plate.	Alg- MAE improved the growth rate of *L. delbruecki* ssp. *bulgaricus* by 75% (at 0.10% (*w/v*) inclusion), 150% (at 0.50% (*w/v*)), 40% (at 0.10% (*w/v*)), and 34% (at 0.30% (*w/v*)) for *L. casei* when compared to the unsupplemented media.	[24]
*Sargassum glaucescens*	Fucoidan	Hair growth-promoting (alopecia treatment)	Cell proliferation: Human follicle dermal papilla cells (HFDPC) in DMEM with 1% FBS for 24 h, then treated with different molecular weight fucoidans (HHP-1-MW, SCW-1, and SCW-5; 1 mg/mL). HFDPC treated by glucose (1 mg/mL) as control; cell viability was measured with CCK-8 assay. Hair follicle culture assay: 5-week-old male C57BL/6 mice. Cultured in William’s medium with or without the supplementation of SCW-5 (1 mg/mL) or minoxidil (1 µM).	Cell proliferation was higher than glucose or PBS treated control. SCW-1 was the most effective with cell viability (200%) of HFDPC. The hair follicles treated by SCW-5 after 69 days of treatment had better hair growth than the minoxidil and control groups (*p* < 0.05).	[75]
*Sargassum angustifolium*	Fucoidan	Wound healing	The wound healing effect of crude fucoidan extracts on adipose-derived mesenchymal stem cells (ADMSCs) was determined by MTT and scratch assays.	The crude extracts of fucoidans were demonstrated by the MTT assay to improve growth up to 1.5 times. With the scratching technique, an increase in cell migration of 76 and 142% was observed after 48 and 72 h of incubation, respectively, in ADMSC cells.	[76]

### 4.2. Antioxidant Activity

Reactive oxygen species (ROS) are involved in biological reactions and intracellular signaling pathways. They are normal products derived from metabolism, such as hydroxyl radicals (OH•), superoxide radicals (•O^2−^), peroxyl radicals (ROO•), peroxide organics (ROOR’), peroxynitrite (ONOO^−^), and hydrogen peroxide (H_2_O_2_). However, a major problem is caused when oxidative stress is present due to the abnormal proliferation of ROS, which has the potential to induce damage in cells in a significant range; vital biomolecules, such as DNA, proteins, and lipids, among others, are affected by oxidative stress and can cause serious diseases, including neurodegeneration diseases, cancer, arthritis, and atherosclerosis [54,65]. Thus, scientists have been searching for years for potential solutions, such as antioxidants that have a significant role in inhibiting oxidation reactions caused by ROS.

Nowadays, it is known that synthetic antioxidants have generally been used in the food industry as additives. Their long-term use produces side effects, bringing notable attention to natural antioxidants. Algal extracts from different species have demonstrated a crucial opportunity to contribute to this sector; in fact, more than fifty species of brown algae from around the globe have been reported to show significant antioxidant activities [77]. The main bioactive compounds with antioxidant activity in brown seaweed species are phlorotannin, fucoxanthin, and polysaccharides, such as alginic acid, fucoidan, and laminarin. These have been studied extensively due to the interest in their potential implementation in pharmaceuticals [78].

Laminarin presents potential antioxidant activity, especially against oxidative stress caused by ROS and free radicals. Crude laminarin extract from brown algae *L. hyperborean* has exhibited higher DPPH radical scavenging (38.62%) compared to commercial laminarin standard (13.93%) from Sigma^TM^ (Sigma-Aldrich, St. Louis, MO, USA). These results agree with the theory that the polysaccharides’ structures have antioxidant activity [79]. In vivo trials have been reported, usually in rats and porcine, to demonstrate the potential effect of laminarin in pulmonary and lipid oxidations, in addition to increased natural antioxidant properties and mitigating ROS generation [18].

Fucoidan antioxidant assays have been applied alone and combined with other sulfated polysaccharides. Fucoidan from *C. Sinuosa* displayed a high antioxidant capacity in 2,2-diphenyl-1-picryl-hydrazyl-hydrate (DPPH) and superoxide dismutase (SOD) assays, showing remarkable scavenging activity of 89% at 750 μg/mL^−1^, DPPH IC_50_ of 46.2, and SOD IC50 23.7 [62]. Sulfated fucoidan of *Sargassum polycitum* has also demonstrated antioxidant behavior using the ferric reducing antioxidant power (FRAP) method; the results of IC_50_ of 41,667 ppm show intense activity [80]. Other recent fucoidan antioxidant results demonstrating the activities of *Sargassum ilicifolium*, *Silvetia Compressa*, and *Sargassum siliquosum* are described in Table 3.

Alginate has been widely used as an antioxidant agent, including as a crude extract, combined, and even as base material for encapsulation. The polysaccharide has indicated an ability to scavenge free radicals and reduce ROS disorder. Favorable antioxidant results of alginate alkaline treatment have been reported in a range from 35.83 to 120.48 μMTEg^−1^ and temperature stable up to 50 °C, decreasing in the temperature range of 70–100 °C [81]. However, recent studies have suggested that alginate oligosaccharides have a better and significant enhancement in antioxidant activities and the capacity to protect endothelial cells, providing a possible therapeutic application for atherosclerosis and related diseases [29].

### 4.3. Anti-Inflammatory Activity

External stimuli, such as injuries or the appearance of pathogenic microorganisms, as well as internal stimuli, such as stress that is considered harmful for any living system, trigger the natural response of inflammation. The inflammatory reaction includes the activation of the immune system, particularly the macrophage and neutrophil cells, which, during the inflammation process, generate secondary factors, such as pro-inflammatory cytokines, nitric oxide (NO), and prostaglandin E2 (PGE2) [82]. The cytokines known as interleukin-1B (IL-1B), interleukin 6 (IL-6), tumor necrosis factor (TNF-alfa), and interleukin-12 (IL-12), as well as enzymes, such as cyclooxygenase (COX-2) and matrix metalloproteinase-9 (MMP-9), are global biomarkers of inflammation [83,84,85].

The biopolymers from brown macroalgae have shown strong anti-inflammatory activities in in vitro and in vivo models. Wang et al. [29] obtained a rich sulfated polysaccharide extract from the *Sargassum fulvellum* and tested its anti-inflammatory properties against RAW 264.7 macrophages and zebrafish embryos stressed by *E. coli* lipopolysaccharides. The extract was able to increase the cell viability of the macrophages up to 94.6%, while the levels of NO, PGE2, TNF-alfa, and interleukins 1 and 6 decreased significantly (Table 3). In a similar way, the zebrafish embryos treated with the sulfated polysaccharides increased their survivability by 70%; in addition, the NO levels, the reactive oxygen species, and the overall cell death was reduced as an effect of the sulfated polysaccharides. Both in vivo and in vitro experiments showed a concentration-dependent behavior, and the best results were obtained with the higher polysaccharide concentration of 100 µg/mL. Another study by Manikandan et al. based on an in vivo model reported high anti-inflammatory activity from a fucoidan extracted from brown algae [69]. They used fucoidan extracted from *Turbinaria decurrens* to explore its activity against formalin-induced paw edema in Swiss albino mice. Their results demonstrate that the polysaccharide administered orally was able to reduce the licking time of the mice of the paw edema by more than 55%, meaning it reduced the mice’s perception of the wound. In addition, the inflammation of the paw edema was reduced by 52% utilizing the standard treatment drug called dexamethasone. An analysis of the paw tissues of the mice reported a reduction in the expression of the COX-2, IL-1B, and MMP-9 genes in comparison to the gene’s expression of the untreated mice with induced paw edema. A study exploring the anti-inflammatory activity of polysaccharides from brown algae was made by Wang et al. [36]: they extracted sulfated polysaccharides from the *Sargassum fusiforme* by an enzymatic method and tested the extracts on LPS-induced stress RAW 264.7 macrophages. The anti-inflammatory activity was analyzed by reading the level of expression of common inflammatory factors, such as NO, TNF-α, IL-1β, IL-6, and PGE2. The sulfated polysaccharides were able to decrease the level of expression of these markers in a dose-dependent behavior; in addition, the cellular viability was enhanced as a result of the reduction of the NO levels.

### 4.4. Antiviral Activity

Viral infections have caused a severe burden on public health around the globe, especially with the pandemic caused by SARS-CoV-2. Viruses are divided into two types: simple (or non-enveloped), which are made up of nucleic acid and a protein capsid, and complex (enveloped), which have a lipoprotein envelope over the protein capsid, making them stronger to protect against environmental factors and conferring protection from viruses to disinfectants or antiviral agents. Nevertheless, to reduce collateral effects from synthetic drugs, pharmacology has been searching for and developing novel, natural, and antiviral agents in order to help alleviate symptoms, shorten the disease period, and minimize side effects and toxicity [86]. Recently, it has been reported in several studies that the antiviral properties of sea life is not only limited to polysaccharides from brown algae, but can also be found in sea cucumber, navicula, green algae, blue–green algae, red algae, and even in chondroitin sulfate from sharks. Recent research indicates the capacity to mitigate viruses by preventing them from interacting with host cells, inhibiting RNA replication and protein synthesis [87].

In this sense, a wide range of studies has been reported on alginate-based materials to verify the antiviral properties of more than 15 types of viruses that can infect different organisms. Alginic acid has been tested against the rabies virus in chicken embryo-related cells, showing a dose-dependent inhibitory effect from 1 to 100 μg/mL. Sodium alginate has also inhibited potato virus X by 95% using Chenopodium quinoa as a host; a strong antiviral effect was demonstrated at a concentration of 1 μg/μL [88]. Furthermore, these previous studies led to research lines for alginate-based biomaterials against SARS-CoV-2.

Laminarin acts by promoting the humoral immune response of virus-infected host cells and activating natural killer (NK) cells and T-lymphocytes [89]. Therefore, laminarin isolated from brown seaweed could be a source of new alternatives against HIV. Shi et al. [90] used a concentration of 50 mg/mL and demonstrated low cytotoxicity; furthermore, they were able to block the adsorption of the virus and suppress the reverse transcriptase. Additionally, other authors have found a positive antiviral response from the laminarin of *L. japonica* against respiratory viruses, such as H5N1 and the RSV virus [71].

Fucoidan is the brown algae biopolymer with the largest spectrum of antiviral properties reported; nevertheless, its bioactive capacity depends on the large size chain, molar composition, and structural attributes, such as the molecular weight and chemical compositions, which can be affected by variations in the season and the specie [91]. There is a wide range of viruses that fucoidan can be inhibit as an antiviral, such as RNA and DNA viruses, including HIV, HSV1-2, ASFV, HTLV-1, MPMV, dengue virus, and cytomegalovirus. Moreover, fucoidan can regulate mitosis or cellular apoptosis due to its capacity to inhibit digestive enzymes and interrupt glucose absorption. It has been shown to have positive responses against HIV. Fucoidan from *Sargassum henslowianum* was used against the herpes virus (HSV-1 and HSV-2) and demonstrated an IC50 of 0.82–0.89 μg/mL by plaque reduction assay; it also showed 0.48 μg/mL against HSV-2 [92]. Recently, the antiviral action of fucoidan has been tested to determine its usefulness against the current pandemic; in fact, in vitro models have demonstrated efficacy against SARS-CoV-2, significantly inhibiting the effect of viral spike protein binding [93]. Fucoidan concentrations between 9.10–15.6 μg/mL have inhibited SARS-CoV-2 in vitro via S glycoprotein binding. In addition, some reports of comparations of different weights of fucoidan have been performed using *Saccharina japonica* in HMW (8.3 μg/mL), significantly better than LMW (16 μg/mL) [94,95].

### 4.5. Non-Conventional Activities

Besides the known variety of studied bioactivities of polysaccharides extracted from brown algae, these polysaccharides have recently been tested in new applications. The latest evidence proves that these polysaccharides are able to benefit the gastrointestinal tract, improve the angiogenesis process, soothe metabolic syndrome, and enhance bone health [96].

The process in which new blood vessels are formed from old blood vessels is known as angiogenesis. This process consists of four stages: (1) the vascular permeability rises, (2) the endothelial cells travel through the extracellular matrix, (3) the differentiation occurs, and finally, (4) the new vessels form and mature after a short time [97]. The failure of any part of this process can lead to metabolic and cardiovascular disorders, but most importantly, can also lead to the growth of carcinogenic cells and therefore to cancer and potential metastasis [98]. Brown macroalgae biopolymers have been used as a countermeasure for uncontrolled angiogenesis. Oliveira et al. [73] used a fucoidan extracted from *Fucus vesiculosus* to prevent the formation of new blood vessels in endothelial cells and in chicken embryos. The extracted fucoidan at a concentration of 0.5 mg/mL was able to prevent the formation of more tubular formations on epithelial cells and the presence of the platelet-derived growth factor (PDGF) was downregulated; this factor is necessary for the proper maturation of blood vessels. In addition, the cells treated with this fucoidan presented a tendency to aggregate instead of spreading and connecting with each other. The chicken embryos treated with the fucoidan presented a decrement of blood vessels and the tumoral mass, supporting the activity of fucoidan as a component to prevent tumor progression. Another similar experiment was conducted by Ohmes et al. [74]. They tested fucoidan extracted from *Fucus distichus* subsp. evanesces into mono- and co-cultured human outgrowth endothelial cells (OEC) and human mesenchymal stem (MSC) cells. The fucoidan was obtained by enzymatic extraction with cellulases and alginate lyases, then different concentrations of fucoidan were tested in the different cultures. The level of expression of different genes, such as angiopoietins-1 (ANG-1), angiopoietins-2 (ANG-2), vascular endothelial growth factor (VEGF), and stromal-derived factor 1 (SDF-1), were measured to analyze the anti-angiogenic properties of the fucoidan extracted, as well as the length and area of the tube-like structures formed by the co-culture of the cells. The fucoidan was able to downregulate the expression of all the genes in MSC; however, the ANG-1 and ANG-2 in the OEC did not decrease because of the fucoidan. The fucoidan extract was able to disrupt and decrease the length and area of the tube-like structures formed during the co-culture of both MSC and OEC cells, showing a strong bioactivity against angiogenesis.

In addition, several researchers have found a prebiotic effect on human intestinal microbiota in brown algae polysaccharides. This is because they are not digestible by hydrolytic enzymes and are fermented in the colon by *Lactobacillus* and *Bifidobacterium*, improving growth and decreasing the concentrations of pathogens. However, digestion affects the activity of algae polysaccharides, therefore it is essential to verify resistance to hydrolysis under in vitro conditions [99]. Okolie et al. [24] worked with sodium alginate extracted from *Ascophyllum nodosum* by different extraction methods. The prebiotic activity was demonstrated by the in vitro growth rate of the *Lactobacillus delbruecki ssp bulgaricus* strain in growth media supplemented at 0.10, 0.30, and 0.50% (*w/v*). It was shown that the activity level depends on the concentration compared to the medium without supplements as a control; it was also shown that there are no significant differences between the techniques of extraction methods. It is important to note that in vivo experiments have generally been performed on rats, pigs, and mice. Zheng et al. [100] reviewed many of these and reported studies for the three brown algae biopolymers that had shown the prebiotic effect, especially by the *Bacteroides* genus, which projects the great potential of these macroalgae biocomposites for functional foods and drugs.

Polysaccharides from brown algae have attracted particular attention in the biomedical field due to their unique properties, such as biocompatibility, biodegradability, non-toxic, non-immunogenic, moisture-retaining, swelling ability, and resembling the structure of the extracellular matrix. Over the last few decades, polysaccharides have been used in biomedical treatments, especially for drug delivery systems, wound healing, and tissue engineering using modern polymer-production technologies, such as 3D-bioprinting or electrospinning [101]. Wound healing consists of several overlapping phases that are intended to restore the anatomical structure and retrieve function of damaged skin. Tissue engineering aims to regenerate damaged tissue/organs using cells, growth factors, and scaffolds [102]. Biopolymers, such as alginate, collagen, and chitosan, are the most used raw material for scaffold manufacturing due to their good plasticity behavior, drug compatibility, and biodegradability; however, synthetic biomaterials improve mechanical properties [103]. Alginate-based biomaterials have been the most used brown algae biopolymer due to their superior capacity to form scaffolding materials, including hydrogels, microcapsules, foams, sponges, and fibers [104]. Iglesias-Metujo and García-Gonzalez designed 3D-printing aerogel scaffolds for bone regeneration in an alginate concentration range of 6–10 wt% and a CaCl_2_ concentration of 0.5 M. In this study, the authors added hydroxyapatite to preserve the geometry of the strands, while the structural stability and yielding scaffolds were improved. The alginate–hydroxyapatite scaffolds were highly porous; furthermore, they were able to attach and proliferate mesenchymal stem cells, in addition to presenting an enhancement of the fibroblast migration in damaged tissue, which supports the bone regeneration potential. There currently exists a wide range of commercial wound dressing products using sodium or calcium alginate in combination with bioactive compounds for specific applications in biomedicine, such as Algicell™, Integra LifeSciences Corp ™, Biatain™, Comfeel Plus™, and Nu-derm™, among others [105].

## 5. Challenges in Research and Marketing of Products Based on Brown Algae Polysaccharides

The emphasis on brown macroalgae is reflected in its worth in the global market, which is stipulated to be USD 16.6 billion in 2020 with a growing rate of 10.8% annually [106]. The global market for fucoidan, alginate, and carrageenan were valued at USD 70.0, 728.4, and 871.7 million, respectively, in 2022, and is expected to continue growing, registering a compound annual growth rate (CAGR) of 11.8, 5.0, and 5.4%, respectively, over the forecast period (2022–2030) [107,108,109].

As explained in the sections above, polysaccharides from brown algae offer important industrial applications, as well as potential benefits for human health. In fact, the food industry makes use of approximately 40% of the total seaweed produced annually, with a value close to 24 million tons, which does not include the macroalgae used as hydrogels and thickeners in the food processing industry [52]. Due to its composition, novel snacks can be easily produced that complement their nutrimental value with proteins, polysaccharides, minerals, and lipids from brown macroalgae, which are also labeled as a vegan type of food [110]. The recent interest in the polysaccharides from brown macroalgae has led to the use of them against several cancer lines, human pathogens, and other microorganisms, and they have proved to be effective against them. Macroalgal preparations are not allowed to be used as medicines since in vivo validation and application studies are still lacking to support these extracts in meeting the various global, national, and local medical regulations [77]. In addition to the conditions that prevent the use of polysaccharides from brown macroalgae as a therapy for diseases is the lack of an efficient drug delivery method with great adsorption and bioavailability.

Fucoidan is the most studied polysaccharide and has the most properties and benefits reported in various studies. Some of its more beneficial properties include anticancer; anti-inflammatory; immunomodulatory effects; protection against neurological diseases, such as Alzheimer’s; and shielding against bacterial and viral infections. Many other studies have examined other benefits, such as its form of application from oral to intravenous and its bioavailability. However, despite several advances, patents, and developments, currently, the only regulatory approval for the use of fucoidan in the U.S. and Europe is for supplements and cosmetics [111].

The numerous biological, antimicrobial, and pharmaceutical properties that fucoidan presents have triggered different strategies for its formulation, release, and application. Different studies have shown different forms of application of fucoidan, from oral administration to inhalable dosage forms and even topical and injectable dosage forms [112].

The oral administration of fucoidan as a powder, tablet, and nanoparticles, which is possible due to its low solubility and low gastric absorption, is often selected when administering it as a gastroprotective dietary supplement, an anti-inflammatory, and a suppressor of oxidative stress to treat gastric ulcers. Fucoidan also has prebiotic effects due to the production of oligosaccharides derived from the degradation by the flora of the gastrointestinal tract, which induces the production of beneficial fatty acids [113].

Even though there is no industrial production of items comprising brown algae that target specific medical applications, there is a wide list of items and patents of products derived from brown algae that cover important needs. Some of the most important applications include an already commercialized crop-stimulating agent that induces a protective response of wheat and durum against a pathogen fungus known as *Zymoseptoria tritici*. Several patents have been reported that explain the process to create organic fertilizers, a flocculant agent, a composite for cosmetic purposes, and a nutritive gel from brown algae. This demonstrates the undiscovered potential of different compounds from brown algae [113].

Besides the limitation caused by normativity, there is a disadvantage regarding the use of the polysaccharides from brown algae: in higher amounts, brown algae has a tendency to absorb heavy metals during its growing process. The *Sargassum* sp. are recognized as a natural biosorbent for pollutants, such as pesticides, mining waste components, oil spills, and heavy metals; therefore, it is necessary to remove the contaminants before extracting the polysaccharides, which makes the use of these biopolymers more expensive [114]. For industrial applications based on the use of a *Sargassum* biomass, the main challenge to establish a process for the extraction of biocompounds derived from *Sargassum* is its the arrival to the coasts due to the temporality of its season; brown algae is only available during certain months of the year, typically between April and October, with great variations in the amounts of macroalgae, which can limit or saturate the harvesting, drying, and processing. Regulations for its collection should be followed in some regions to avoid changes in the environmental dynamics and because *Sargassum* is treated as a protected natural resource. [115].

Estimating the seasonal arrivals to the impacted regions by monitoring technologies through remote sensors, such as satellite information, has been reported, but these technologies present limitations, such as the loss of information on the movement of *Sargassum* due to the presence of clouds, which can last for days. An additional limitation includes spatial resolution, since only small areas are monitored. It should also be borne in mind that the main complication is that *Sargassum* is a living organism that interacts and responds to a highly changing and complex environment, just as sea currents generate variations in the arrival of *Sargassum* to different coastlines [116].

Currently, *Sargassum* is little used compared to the ton that arrive on beaches every year in several regions along the Atlantic Ocean. Despite all the efforts to explore its potential applications and to enhance its security, there are still opportunities to engage leaders from various sectors to promote the use of brown algae to accomplish a common goal. For example, government entities, industry experts, and researchers could face the environmental problems derived from *Sargassum* together. Finally, the extraction of high value compounds, the limitations of harvesting and storing large quantities of seaweed, and the processing of *Sargassum*, as well as planning for occasions when it does not arrive are all factors that need to be considered [117]. Although the macroalgae market already represents a very high economic volume, its use is based on biopolymers, such as alginates, laminarin, fucoidan, and other compounds. This shows that the full potential of macroalgae is not being fully exploited as an integral system of a biorefinery to take advantage of all the valuable compounds, such as phenolic compounds, pigments, different polymers, or minerals, nor is biochar produced with the residue of the previous processes.

## 6. Conclusions

The current stress on the environment to obtain natural resources for food use, biomaterials, medicines, and many other applications, has resulted in catastrophic consequences for the environment and all living beings due to contamination, increased incidence of diseases, new viruses and pandemics, and many other problems that we may be unaware of today. Valorizing natural resources that abound in ecosystems, such as macroalgae, without interfering or causing damage to the ecosystems in which they are found can be the solution to generate new products, such as food, medicines, and biomaterials with low environmental impact. *Sargassum* has presented a great environmental problem due to its increasing presence on the coasts, causing the death of various aquatic species, along with serious tourism and human health problems, resulting in great economic losses worldwide. Therefore, to reverse the negative impact of this brown macroalgae, its application has been sought in various fields, particularly in the medical field, due to its large number of biocompounds, consisting mainly of polysaccharides, which have been shown to contain diverse therapeutic properties to treat cancer and immune system diseases, regulate blood pressure regulation and cholesterol, and act as probiotics. The extraction methods for obtaining the polysaccharides, namely, SFE, SCWE, UAE, EAE, and MAE, are considered green technologies because they make less of an impact on the environment and use of organic solvents. The yields of bioactive compounds depend on the extraction method used and the conditions, as well as the environmental conditions, season, temperature, and location, causing large variations in compound yields and bioactivities. Therefore, the products obtained from *Sargassum* will vary between batches according to these factors.

Food derived from *Sargassum* is currently on the market due to its high nutritional value; it is also available as aquafeed and food for pets. In the therapeutic area, even with many scientific articles and patents demonstrating the bioactivities and different applications of *Sargassum* extracts and compounds, such as polymers, more studies are required to validate the stability of these formulations and their safety for human consumption. However, many nutraceutical or functional products based on *Sargassum* and other brown algae are marketed as alternative medicine for the treatment of certain diseases or ailments. In the near future, ocean resources may be utilized as sustainable, renewable, and abundant natural resources that can be exploited as a source of food, medicines, and supplies for the most essential human needs.

## Figures and Tables

**Figure 1 plants-12-02445-f001:**
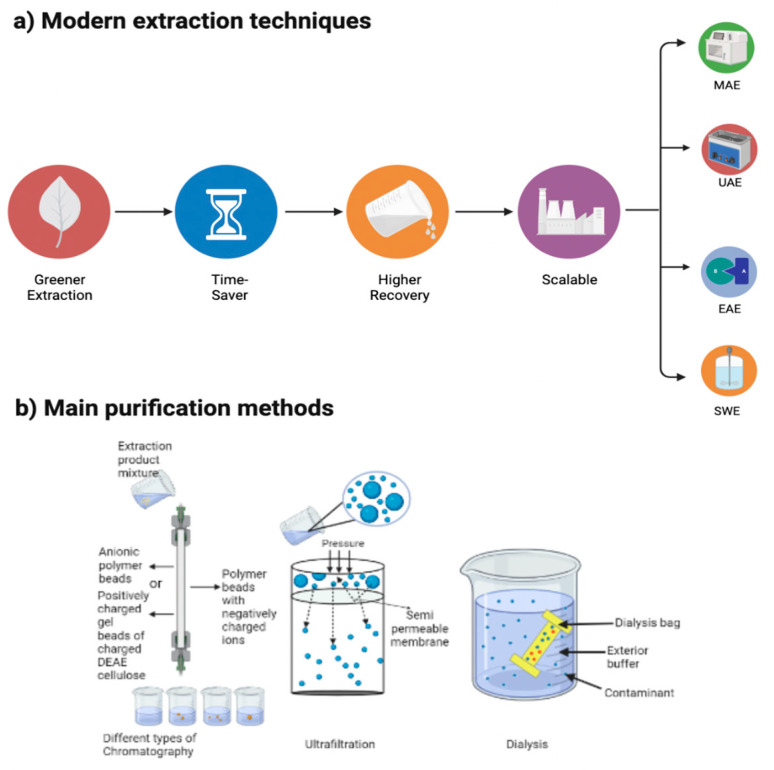
(**a**) Novel extraction techniques and main purification methods for biocompound recovery. MAE—microwave-assisted extraction; UAE—ultrasound-assisted extraction; EAE—enzyme-assisted extraction; and SCWE—subcritical water extraction; (**b**) Purification methods: Chromatography, Ultrafiltration and Dialysis. Created with BioRender.com and extracted under premium membership.

**Figure 2 plants-12-02445-f002:**
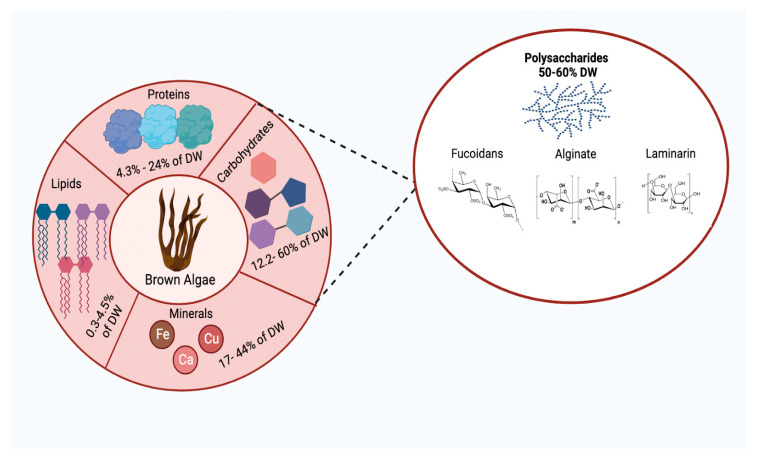
Scheme showing the composition of brown algae, with carbohydrates representing the highest percentage (12.2–60% of DW). Within this group are polysaccharides, which are of great interest due to their therapeutic properties, including laminarin (22–49% DW), fucoidans (5–38% of the DW), and alginate (40% DW). Created with BioRender.com and extracted under premium membership.

**Figure 3 plants-12-02445-f003:**
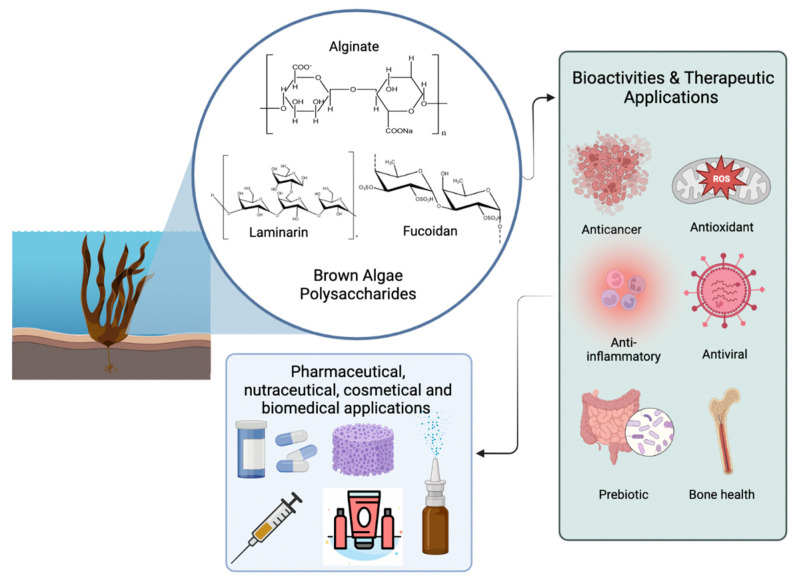
Bioactivities and therapeutic perspectives of brown algae polysaccharides. Created with BioRender.com and extracted under premium membership.

**Table 1 plants-12-02445-t001:** Recently used methods of extraction of polysaccharides from several brown algae species.

Algae	Polysaccharide	Pre-Treatment	Extraction Method	Extraction Details	Recovery	References
*Sargassum siliquosum*	Fucoidan	Washed with (H2O); dried (60 °C) for 48 h; powdered with single-shaft extruder under conditions of 115 °C, 10 kg/h. 360 rpm; sieved with 20 mesh sieves.	MAE & UAE	1 g powder Solvent: EtOH 10 mL/g 25 °C for 4 h. MAE: 750 W; 10 min; 15 mL/g UAE: 100 W; 10 min; 15 mL/g	MAE 6.94% UAE4.78%	[29]
*Fucus vesiculosus* *Fucus serratus Fucus evanescens*	Laminarin & Fucoidan	Grind-dried seaweed, washed with EtOH and acetone.	MAE	1.5 g Solvent: 25 mL sulfuric acid [10 mM] 120 °C for 30 min. Precipitation of laminarin EtOH (40% *v*/*v*); precipitation of fucoidan EtOH (70% *v*/*v*)	Laminarin 8.68% Fucoidan 5.56%	[30]
*Ascophyllum nodosum*	Fucoidan	80% EtOH; 20 h; room temp. 80% EtOH; 5 h; 70 °C.	UAE	0.01 M HCl, 35 min, 40% amplitude; 20 kHz 2% CaCl_2_, overnight; 4 °C.	4.56%	[22]
*Nizamuddinia zanardinii*	Fucoidan	Washed, dried (40 °C), milled, and stored in the freezer. (50 g) suspended in 500 mL of 85% EtOH, stirred (24 h, 25 °C), rinsed with acetone, and dried under laminar hood (22 ± 2 °C).	EAE & UAE & EUAE (Alcalase)	EAE (2.5 mL/dry material weight, pH 7, solid-to-solvent ratios 1:30 g/mL) for 24 h at 50 °C. UAE sonicated with distilled water (1:76 g/mL) with (frequency 20 kHz, max power 400 W, Ø = 1.3 cm) at 196 and 70 °C for 59 min. EUAE (2.5 mL/dry material weight, pH 7, temperature 50 °C, solid-to-solvent ratios 1:30 g/mL) for 23 h. Sonication (196 W, 70 °C, 59 min)	EAE 5.58% UAE 3.6% EUAE 7.87%	[31]
*Saccharina Japonica*	Fucoidan	Washed, chopped, freeze-dried (−80 °C. 72 h), ground. Samples that passed through a 710 μm sieving mesh were used.	SCWE	200 cm^3^ batch system 0.1% NaOH 80 bars S/L ratio 0.05 g mL^−1^, 127.01 °C, 300 rpm, 11.98 min	13.56%	[32]
*Nizamuddinia zanardinii*	Fucoidan	Washed with water, dried at 40 °C for 72 h, sieved (<0.5 mm).	SCWE	29 min, 150 °C. Raw material to water ratio 21 g (mL)	25.98%	[31]
*Ascophyllum nodosum*	Fucoidan	Oven-dried (50 °C). Ground for 9 days (1 mm particle size). Maceration 10 min, 0.1 M HCl, room temperature.	UMAE	Sonication: 50 W, 20 kHz, 100% ultrasonic amplitude. Microwave 2450 MHz 1000 W, 5 min.	3.53%	[33]
*Nizamuddinia zanardinii*	Fucoidan	Cleaned, rinsed on the spot with seawater. Washed with distilled water and oven-dried at 40 °C for 72 h. Powdered.	Alcalase Flavourzyme Celluclast Viscozyme	(5% *v*/*v*, pH 8, 50 °C, 24 h). (5% *v*/*v*, pH 7.5, 50 °C, 24 h). (5% *w*/*v*, pH 4.5, 50 °C, 24 h). (5% *v*/*v*, pH 4.5, 50 °C, 24 h). Boiled at 95 °C 15 min and cooled in ice bath. Centrifugation (10 min at 9000 rpm)	5.58% 4.36% 4.80% 4.28%	[31]
*Sargassum muticum*	Alginate	Washed with H_2_O.	UAE	*v*/*m* 20:1 (wt) Solvent: H_2_O; 25 °C, 30 min, 1.5 A, 50 W, 40 Hz.	15%	[34]
*Sargassum binderi*	Alginates	Washed with H_2_O, dried, milled with a blender. Stored under vacuum. EtOH treatment for dried seaweed (overnight) 25 °C. Filtered with 10 µL Millipore nylon mesh. Washed with distilled water.	UAE	10 g/L Solvent: H_2_O pH: 11 150 W 30 min 90 °C. 25 kHz	27%	[35]

MAE—microwave-assisted extraction; UAE—ultrasound-assisted extraction; EAE—enzyme-assisted extraction; EUAE—enzyme-ultrasonic-assisted extraction; SCFE—supercritical fluid extraction; SCWE—subcritical water extraction.

## Data Availability

Not applicable.

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
