# Peer review of "Polysaccharides from the Sargassum and Brown Algae Genus: Extraction, Purification, and Their Potential Therapeutic Applications"

_plants, 2023, doi:10.3390/plants12132445_

Round 1

Reviewer 1 Report

Authors performed a thorough review on the extraction, purification and applications of biopolymers from Brown algae. The manuscript fits well with the scope of the journal. I recommend the manuscript to be accepted by addressing some minor issues below:

1. Authors have mentioned that ecosystem issues of high abundance of Brown algae is the one of the driving forces for developing therapeutic applications of them. There might be heavy metal or other toxins enrichment of brown algae from the in environment. In terms of the purification and extraction of these biopolymers, do people discuss the purity and amount of toxins left after the purification? Can authors add some discussion about this?

2. In the section of anticancer activity of those polysaccharides, most of anticancer activities were conducted from the cell-based in vitro assays. Can authors comment on the cytotoxicity effects of those polysaccharides?

Minor editorial issues:

1. The format of some references, such as journal names abbreviation format, isn't consistent. 

2. There is too much empty space in table 2. It can be removed to make the table fit better in the page. 

Author Response

Please find below the detailed responses to the Reviewer’s/Editor’s comments. All changes/corrections made have been highlighted with GREEN background in color. We have also enclosed the thoroughly revised version of the manuscript mentioned above according to the Reviewer’s comments, suggestions, and recommendations. 

Reviewer 1

Authors performed a thorough review on the extraction, purification and applications of biopolymers from Brown algae. The manuscript fits well with the scope of the journal. I recommend the manuscript to be accepted by addressing some minor issues below:

  1. Authors have mentioned that ecosystem issues of high abundance of Brown algae is the one of the driving forces for developing therapeutic applications of them. There might be heavy metal or other toxins enrichment of brown algae from the in environment. In terms of the purification and extraction of these biopolymers, do people discuss the purity and amount of toxins left after the purification? Can authors add some discussion about this?

 Author’s Response:Thank you for the observation. The subject of heavy metals is addressed in the manuscript, but experience in the use and application of different macroalgae in products and foods has shown us that the presence of these heavy metals is low, and the processes of washing, extraction, precipitation, and Other processes greatly reduce the presence of contaminants, as well as the incorporation of the extracted compounds in finished products is always done in a low proportion, resulting in the presence of these contaminants being almost imperceptible and well below the regulations. 

  1. In the section of anticancer activity of those polysaccharides, most of anticancer activities were conducted from the cell-based in vitro assays. Can authors comment on the cytotoxicity effects of those polysaccharides?

 Author’s Response:Thank you for the commentary, the main mechanism is by inducing cell apoptosis and inhibiting angiogenesis. The information was added.

Minor editorial issues:

  1. The format of some references, such as journal names abbreviation format, isn't consistent. 

Author’s Response:Thank you for the observation, it was revised and modified.

  1. There is too much empty space in table 2. It can be removed to make the table fit better on the page.

Author’s Response: Thank you for the observation, it was modified.

Reviewer 2 Report

The review article, "Brown algal polysaccharides: extraction, purification, and their potential therapeutic applications", describes components of brown macroalgae (polysaccharides), their purification methods and application. The goals and motivation of such a study are relatively straightforward. After considering the issues outlined below, the rewritten paper can contribute to Plants.

Additional comments that need to be addressed:

1)     Although the manuscript is written well enough in English to be understood, the English language and style must be carefully revised throughout the whole manuscript. Some suggested revisions are: 

-        Many sentences have extra spaces between punctuation marks and words.

-        Sargassum was written sometimes in lowercase and sometimes in uppercase. Please standardize this throughout the manuscript.

-        The sentence: As for lyophilization or freeze drying consists in eliminating the water from the sample of interest, through of freezing the water, afterwards the frozen water is removed by vacuum, by passing the matter from the solid state to vapor (via sublimation), and the unfrozen water is eliminated by desorption [41].' should be Lyophilization or freeze drying consists in eliminating the water from the sample of interest by freezing the water, afterwards, the frozen water is removed by vacuum, bypassing the matter from the solid state to vapour (via sublimation), and the unfrozen water is eliminated by desorption [41].

-        The sentence: 'While ultrafiltration promote the separation of materials such as fats, particles, proteins, among others of the sample of interest, through the use of membranes that can have a pore size 180 of 0.01 or 0.1 μm, and using a pressure ranging from 3.4 to 8.3 bar [42].' should be 'While ultrafiltration promotes the separation of materials such as fats, particles, and proteins, among others of the sample of interest, through the use of membranes that can have a pore size 180 of 0.01 or 0.1 μm and using a pressure ranging from 3.4 to 8.3 bar [42].'

2)     Please explain the abbreviation DEAE.

3)     In each chapter should be written short conclusion and the most important findings.

4)     It would be worth writing a deeper comparison and showing the relationship between the extraction conditions, such as the extraction times and temperatures, and kinds of solvents etc., brown macroalgae based on the extraction of diverse polysaccharides such as fucoidans, alginates and laminarin etc.

5)     The conclusion is too general and too short.

Moderate editing of English language required

Author Response

Please find below the detailed responses to the Reviewer’s/Editor’s comments. All changes/corrections made have been highlighted with GREEN background in color. We have also enclosed the thoroughly revised version of the manuscript mentioned above according to the Reviewer’s comments, suggestions, and recommendations. 

Reviewer 2

The review article, "Brown algal polysaccharides: extraction, purification, and their potential therapeutic applications", describes components of brown macroalgae (polysaccharides), their purification methods and application. The goals and motivation of such a study are relatively straightforward. After considering the issues outlined below, the rewritten paper can contribute to Plants.

Additional comments that need to be addressed:

1)     Although the manuscript is written well enough in English to be understood, the English language and style must be carefully revised throughout the whole manuscript. Some suggested revisions are: 

-        Many sentences have extra spaces between punctuation marks and words.

Author’s Response:Thank you for the observation, the document was revised in detail.

-        Sargassum was written sometimes in lowercase and sometimes in uppercase. Please standardize this throughout the manuscript.

Author’s Response:Thank you for the observation, it was standardized.

-        The sentence: As for lyophilization or freeze drying consists in eliminating the water from the sample of interest, through of freezing the water, afterwards the frozen water is removed by vacuum, by passing the matter from the solid state to vapor (via sublimation), and the unfrozen water is eliminated by desorption [41].' should be Lyophilization or freeze drying consists in eliminating the water from the sample of interest by freezing the water, afterwards, the frozen water is removed by vacuum, bypassing the matter from the solid state to vapour (via sublimation), and the unfrozen water is eliminated by desorption [41].

Author’s Response: Thank you for the observation, was modified.

-        The sentence: 'While ultrafiltration promote the separation of materials such as fats, particles, proteins, among others of the sample of interest, through the use of membranes that can have a pore size 180 of 0.01 or 0.1 μm, and using a pressure ranging from 3.4 to 8.3 bar [42].' should be 'While ultrafiltration promotes the separation of materials such as fats, particles, and proteins, among others of the sample of interest, through the use of membranes that can have a pore size 180 of 0.01 or 0.1 μm and using a pressure ranging from 3.4 to 8.3 bar [42].'

Author’s Response: Thank you for the observation, was modified.

2)     Please explain the abbreviation DEAE.

Author’s Response: Thank you for the observation, the abbreviation was added.

3)     In each chapter should be written short conclusion and the most important findings.

Author’s Response: Many thanks for the observation, we revised and added the conclusion in section 2 and 5. In section 3 and 4 we consider that is not necessary. 

4)     It would be worth writing a deeper comparison and showing the relationship between the extraction conditions, such as the extraction times and temperatures, and kinds of solvents etc., brown macroalgae based on the extraction of diverse polysaccharides such as fucoidans, alginates and laminarin etc.

Author’s Response: Many thanks for the comentary, this information is shown in Table 1, including other extraction conditions.However, it is difficult to establish the effect on polysaccharides, such as molecular weight or other properties, since each treatment or extraction technology generates a different effect, which was not addressed in this manuscript.

 5)     The conclusion is too general and too short.

Author’s Response:Many thanks for the observation, the conclusion was improved.

Reviewer 3 Report

I have assessed the review entitled ‘‘Brown algal polysaccharides: extraction, purification, and their potential therapeutic applications’’ by Flores-Contreras et al. Though the title is broad and initially sparks a reader’s interest in the article, the authors have focused on a single/limited number of genera of brown algae (i.e., Sargassum species). In practice, brown algae (or Phaeophyceae) consist of more than 1,500 species (from Ectocarpus, Fucus, Macrocystis, Sargassum etc genera) which vary greatly in their sizes, shapes and colours, depending on the ratio of chlorophyll and fucoxanthin. Surprisingly, Macrocystis genus is not considered at all in this review, yet it contains the largest number of brown algae. In addition, the aspects examined in this review have already been worked on in previous studies with some members from this genus (e.g., Biparva et al., 2023). Taken together, brown algae, have received considerable attention globally, and therefore some reviews have already addressed a fraction of the items highlighted in the current submission (for example, Nigam et al. 2022; Jönsson et al., 2020). This being said, any further review on this topic should bring new perspectives, broaden the context, identify some gaps and come to better conclusions that could drive the field further.

CITED LITERATURE

Biparva et al. (2023). Advanced Processing of Giant Kelp (Macrocystis pyrifera) for Protein Extraction and Generation of Hydrolysates with Anti-Hypertensive and Antioxidant Activities In Vitro and the Thermal/Ionic Stability of These Compounds. Antioxidants, 12, 775. https://doi.org/10.3390/antiox12030775

Nigam et al. (2022). Perspective on the Therapeutic Applications of Algal Polysaccharides. Journal of polymers and the environment, 30(3), 785–809. https://doi.org/10.1007/s10924-021-02231-1

Jönsson et al. (2020) Extraction and Modification of Macroalgal Polysaccharides for Current and Next-Generation Applications. Molecules 25, 930. https://doi.org/10.3390/molecules25040930

Needs some minor corrections which can be addresssed by MDPI in-house prepublication English proofreading team 

Author Response

Reviewer 3

I have assessed the review entitled ‘‘Brown algal polysaccharides: extraction, purification, and their potential therapeutic applications’’ by Flores-Contreras et al. Though the title is broad and initially sparks a reader’s interest in the article, the authors have focused on a single/limited number of genera of brown algae (i.e., Sargassum species). In practice, brown algae (or Phaeophyceae) consist of more than 1,500 species (from EctocarpusFucusMacrocystisSargassum etc genera) which vary greatly in their sizes, shapes and colours, depending on the ratio of chlorophyll and fucoxanthin. Surprisingly, Macrocystis genus is not considered at all in this review, yet it contains the largest number of brown algae. In addition, the aspects examined in this review have already been worked on in previous studies with some members from this genus (e.g., Biparva et al., 2023). Taken together, brown algae, have received considerable attention globally, and therefore some reviews have already addressed a fraction of the items highlighted in the current submission (for example, Nigam et al. 2022; Jönsson et al., 2020). This being said, any further review on this topic should bring new perspectives, broaden the context, identify some gaps and come to better conclusions that could drive the field further.

CITED LITERATURE

Biparva et al. (2023). Advanced Processing of Giant Kelp (Macrocystis pyrifera) for Protein Extraction and Generation of Hydrolysates with Anti-Hypertensive and Antioxidant Activities In Vitro and the Thermal/Ionic Stability of These Compounds. Antioxidants, 12, 775. https://doi.org/10.3390/antiox12030775

Nigam et al. (2022). Perspective on the Therapeutic Applications of Algal Polysaccharides. Journal of polymers and the environment, 30(3), 785–809. https://doi.org/10.1007/s10924-021-02231-1

Jönsson et al. (2020) Extraction and Modification of Macroalgal Polysaccharides for Current and Next-Generation Applications. Molecules 25, 930. https://doi.org/10.3390/molecules25040930

Needs some minor corrections which can be addresssed by MDPI in-house prepublication English proofreading team

Author’s Response: Many thanks for your commentaries and observations. The manuscript has a bit of an impact on sargassum, derived from the global problem of arrivals that every year reach the Caribbean and African coasts, affecting various countries, causing many problems in environmental, economic, social and even tourism terms.

Round 2

Reviewer 2 Report

The authors provided reliable and comprehensive answers. I think that this revised review is suitable for publication without further revision.

Author Response

Thank you for your favorable comments for the publication of our manuscript.

Reviewer 3 Report

Since the authors argue that this submission has ''a bit of an impact on sargassum'', it will be nice to narrow the title and the content to this genus. The current title is of course be misleading then. In this perspective, you should already omit all the other species which are not from the same genus from the review.

Needs to be checked for grammatical fixes and English language 

Author Response

Thank you very much for your comment, we made the modification of the title of the manuscript.